# Patient safety climate in general public hospitals in China: differences associated with department and job type based on a cross-sectional survey

Ping Zhou,[1,2] Fei Bai,[3] Hui-qin Tang,[4] Jie Bai,[1,2] Min-qi Li,[1,2] Di Xue[1,2]

[1]Key Laboratory of Health Technology Assessment, NHFPC (Fudan University), Shanghai, China
[2]Department of Hospital Management, School of Public Health, Fudan University, Shanghai, China
[3]Department of Hospital management, National Center for Medical Service Administration, Beijing, China
[4]Department of Hospital management, Health and Family Planning Commission of Hubei Province, Wuhan, China

**Correspondence to**
Professor Di Xue;
xuedi@shmu.edu.cn

## ABSTRACT

**Objective** This study analysed differences in the perceived patient safety climate among different working departments and job types in public general hospitals in China.

**Design** Cross-sectional survey.

**Setting** Eighteen tertiary hospitals and 36 secondary hospitals from 10 areas in Shanghai, Hubei Province and Gansu Province, China.

**Participants** Overall, 4753 staff, including physicians, nurses, medical technicians and managers, were recruited from March to June 2015.

**Main outcome measure** The Patient Safety Climate in Healthcare Organisations (PSCHO) tool and the percentages of 'problematic responses' (PPRs) were used as outcome measures. Multivariable two-level random intercept models were applied in the analysis.

**Results** A total of 4121 valid questionnaires were collected. Perceptions regarding the patient safety climate varied among departments and job types. Physicians responded with relatively more negative evaluations of 'organisational resources for safety', 'unit recognition and support for safety efforts', 'psychological safety', 'problem responsiveness' and overall safety climate. Paediatrics departments, intensive care units, emergency departments and clinical auxiliary departments require more attention. The PPRs for 'fear of blame and punishment' were universally significantly high, and the PPRs for 'fear of shame' and 'provision of safe care' were remarkably high, especially in some departments. Departmental differences across all dimensions and the overall safety climate primarily depended on job type.

**Conclusions** The differences suggest that strategies and measures for improving the patient safety climate should be tailored by working department and job type.

## INTRODUCTION

Patient safety is a core issue in healthcare services. Both hospitals and the Chinese government have expended substantial efforts to strengthen patient safety climate and improve patient safety performance.[1–4]

Because patient safety climate is associated with positive outcomes such as greater error reporting,[5] fewer adverse events,[6 7] lower

### Strengths and limitations of this study

► This study was conducted in Shanghai Municipality, Hubei Province and Gansu Province, which represent high, middle and low socioeconomic status levels in the Eastern, Central and Western regions of China, using a large valid sample of 4121 respondents from public general hospitals.

► This study is the first to investigate variations in the perceived patient safety climate among different departments and job types and their interaction in China's public general hospitals.

► To depict differences by job type within selected working departments, we incorporated the interaction of the working department and job type variables in the model and graphically displayed the predictions using a heat map.

► Although our analyses represent an important advance over prior studies because we adjusted for important known individual and hospital characteristics, other characteristics that were not measured could play a role in distinguishing personnel by working department and job type.

► The results from 54 public general hospitals in three regions might not be generalisable to all hospitals in China, although our sample size was large and represented public hospitals in high, middle and low socioeconomic level regions in the Eastern, Central and Western regions of China.

mortality rates[8] and lower readmission rates,[9] measuring patient safety climate and understanding its variations can be helpful for targeting efforts to improve patient safety.[10–12] However, patient safety climate can vary within organisations. Previous studies have indicated that the patient safety climate of particular departments varies both across and within institutions.[13–17] Patient safety climate at the unit level can mask important local variations, and measuring individual departments' patient safety climate can identify important opportunities for improvement.

The previous literature has suggested that variations in patient safety climate may be

related to the pace and complexity of the work performed in different work areas.[15 16] Most studies investigating unit-specific climates have focused on measuring the climate in one or more types of departments with higher levels of intrinsic risk, such as the operating room (OR), the intensive care unit (ICU) and/or the emergency department (ED).[13–16 18] However, research concerning the patient safety climates of other units, such as the paediatric, internal medicine, surgery and clinical auxiliary departments (CADs), is insufficient.

Some studies have measured the perception of the patient safety climate among personnel by job type.[14 19–22] In some studies, physicians demonstrated more positive perceptions of the patient safety climate than did nurses and other clinical personnel.[16 18 21] However, in a previous study conducted in hospitals in Pudong New Area, Shanghai (a Chinese municipality), we found that physicians had more negative perceptions of the patient safety climate than did nurses.[22]

Given the variations in the perception of the patient safety climate among different departments and job types, efforts to improve patient safety climate should not be limited to interventions at the hospital level but should extend to the department level and different types of employees. Few articles have focused on patient safety climates at the department level and among different types of employees in hospitals in China.

This study analysed differences in the perception of the patient safety climate among different working departments and job types in 54 public hospitals located in the Eastern, Central and Western regions of China. We selected not only intrinsically hazardous departments but also other departments, including the internal medicine, surgery, obstetrics and gynaecology, paediatric and CADs. We specifically explored (1) in which departments the staff perceived the patient safety climate more negatively, (2) in which job types the staff gave lower scores across safety climate dimensions and (3) whether there were differences by job type in the selected working departments.

## METHODS

### Survey instrument

In the study, we applied the Patient Safety Climate in Healthcare Organisations (PSCHO) tool[23–25] to measure patient safety climate. Although various instruments are available to measure a hospital's safety climate,[26] we selected the PSCHO tool because of its good reliability and validity.[22 24 25] Moreover, it includes 'fear of blame' and 'fear of shame' to measure potential barriers to improving patient safety,[27] both of which capture underlying characteristics of Chinese culture.[22 28 29]

The PSCHO contains 12 dimensions and four categories (based on hospital, work unit, interpersonal contributions to the safety climate and other aspects of the safety climate).[23 24] The PSCHO items use a five-point Likert scale ranging from 'strongly disagree' to 'strongly agree', with a neutral midpoint.

In this study, we added two items to the PSCHO: 'Staff can freely voice their opinions on patient safety' in the dimension 'Psychological Safety', and 'We analyze accidents or unexpected events in a timely manner' in the dimension 'Problem Responsiveness'. These items were added because they reflect a more general psychological safety climate (not specific to certain concerns) and timely responses to adverse events in hospitals, respectively. The survey also asked informants to provide demographic information, including gender, age, education, working years, monthly income, working department and job type.

### Sample

A cross-sectional survey was conducted using a stratified sampling method in Shanghai Municipality, Hubei Province and Gansu Province in China from March to June of 2015. The regions were selected to capture various socioeconomic statuses and geographic distributions within China. First, we selected three provinces/municipalities representing high, middle and low socioeconomic status levels located in the Eastern, Central and Western regions of China.

Hubei Province has 12 prefecture-level cities and one autonomous prefecture. Gansu Province has 12 prefecture-level cities and two autonomous prefectures. We selected three prefecture-level cities or autonomous prefectures (areas) in both Hubei Province and Gansu Province to represent high, middle and low socioeconomic status levels within each province. In each area, two tertiary public general hospitals and four secondary public general hospitals were selected.

Shanghai Municipality has 16 districts. Because Shanghai's tertiary hospitals are not evenly distributed among these districts, six tertiary public general hospitals were selected to represent tertiary hospitals owned by universities, the Shanghai government, or district governments. Additionally, 12 secondary public general hospitals were selected from four districts (A–D) (areas). Because District A is the largest district in Shanghai and comprises both urban and rural areas covering approximately 20% of Shanghai's total population, six secondary public general hospitals were selected in this district. In the other three districts in Shanghai, two secondary public general hospitals were selected.

The sampled public hospitals, including 18 tertiary hospitals and 36 secondary hospitals, represent public hospitals in China quite well, with different numbers of beds (90–3283 beds), at different levels (secondary vs tertiary), from the Eastern, Central and Western regions, and from provinces and areas with different social and economic statuses.

### Data sources

For each selected hospital, general data (ie, hospital level, number of beds, number of physicians and number of nurses) were collected, and anonymous, paper based and self-administered employee questionnaires were

distributed and collected by trained coordinators in the surveyed hospitals or regions according to our study design.

In the employee surveys, we randomly sampled 10% of the managers and administrative staff (at least 15), 10% of the frontline physicians (at least 15), nurses (at least 15) and health technicians and staff working in medical auxiliary departments (at least 5) in each hospital. The term manager refers to a hospital or department director (including working departments and administrative offices), and the term administrative staff refers to non-managerial employees working in administrative offices related to patient safety and medical quality. Frontline workers are non-managerial employees who interact directly with patients. Employees working in internal medicine, surgery, obstetrics and gynaecology, paediatrics, the ICU, the ED, anaesthesiology and OR, CADs and other departments (including stomatology, dermatology, ear, nose and throat, ophthalmology, psychiatry, traditional Chinese medicine departments, administrative and supporting departments) were recruited. The term 'clinical auxiliary departments' refers to laboratory departments, imaging departments, ECG departments, pathology departments, pharmacy departments, supply centres, etc.

Written informed consent was exempted because the questionnaire survey of employees was anonymous and had less than minimum risk. During the survey, anonymous and self-administered questionnaires were distributed and collected according to our study design by trained coordinators who were employees in the surveyed hospitals or administrators from the local health bureaus. Sampled employees who were willing to participate in our study filled out the questionnaires and returned them to the coordinators.

### Survey data analysis
#### Psychometric analysis
We conducted item analyses and confirmatory factor analysis to test the reliability and validity of the PSCHO revised for the Chinese context. In the confirmatory factor analysis, we used PROC CALIS in SAS (SAS Institute) to form a structural equation model with the raw data. The analysis showed that 11 dimensions had high internal consistency (Cronbach's α coefficients ranged from 0.77 to 0.93), but one dimension ('fear of blame and punishment') had a lower Cronbach's α coefficient (0.66). The overall scale had a high Cronbach's α coefficient (0.96). The standardised root mean square residual and the root mean square error of approximation of the PSCHO were 0.049 and 0.058, respectively. The Bentler's comparative fit index, the Bentler-Bonett normed fit index and the non-normed index values were 0.913, 0.906 and 0.907, respectively. The adjusted goodness of fit was 0.83. However, the goodness of fit index (GFI) was 0.84, which is slightly lower than the criterion for this index (GFI >0.85). Overall, the constructive validity of the PSCHO revised for the Chinese cultural context in this study was acceptable.[30–32]

### Statistical analysis
We used the percentage of 'problematic responses' (PPRs) to measure patient safety climate. A rating of <3 for a positive statement or >3 for a negative statement was identified as a problematic response. A lower PPR is indicative of a better perception of the safety climate. This scoring method identifies areas of non-uniformity in safety that are of potential concern and may benefit from interventions to improve the safety climate.[21 22 33–35]

We computed the PPR for each safety climate dimension, with each item in the dimension weighted equally. We also calculated the average PPR for all questions in the survey as a summary statistic, which we referred to as the 'overall safety climate'. These percentages were calculated as the averages of all responses received. Comparisons among working departments and job types were calculated separately for respondents who indicated that they worked in any of the nine types of working departments (internal medicine, surgery, obstetrics and gynaecology, paediatrics, ICU, ED, anaesthesiology and OR, CAD and others) and then for employees by different job type (frontline physicians, frontline nurses, medical technicians, managers and others).

A two-level random intercept model was used to examine how working department and job type affected each dimension and the overall patient safety climate, with the hospital as the level 2 cluster. All models controlled for other respondent characteristics (gender, age, education, number of working years and monthly income) and hospital characteristics (region, hospital level, bed size and doctor–nurse ratio). When estimating the predicted PPR values for each dimension and overall by working department and job type, mean values and proportions were used for continuous covariates and categorical covariates, respectively.

To test the appropriateness of using a two-level model to account for the nesting of individuals within hospitals, we first calculated and tested the intraclass correlation coefficient of the empty model, which included no independent variables for each dimension and the overall PPR.[36] The results revealed significant differences in employees' perceptions of the patient safety climate among hospitals (p<0.0001).

To depict differences by job type (focusing on physicians with various professional titles and other employees) within selected working departments, we added the interaction of the working department and job type variables to the models. To provide an immediate visual summary of the relevant predictions, we applied a heat map that provided a two-dimensional representation of the data using colours. We focused on physicians in the interaction analysis because they are the most important staff members in healthcare services, and we intended to investigate whether there were differences among chief physicians, attending physicians, residents or below and other personnel within a department.

Statistical analyses and graphics were generated using SAS V.9.20, Excel 2007 and R software V.3.3.3.

## RESULTS

### Respondent characteristics

In this study, 4176 staff members from 54 selected hospitals responded to the survey. The response rate and the valid response rate were 87.86% and 86.70%, respectively. Among the valid respondents, 55.67% worked in secondary hospitals, 50.81% worked in Shanghai hospitals and 52.15% worked in hospitals with a large number of beds (>800 beds). The percentages of respondents working in the internal medicine, surgery, obstetrics and gynaecology, paediatric, ICU, ED, anaesthesiology, OR and CADs were 19.46%, 18.99%, 6.54%, 3.15%, 3.59%, 6.08%, 1.55%, 2.21% and 13.51%, respectively. By job type, 38.66% were frontline physicians (including 10.32% associate chief physicians or chief physicians, 15.53% attending physicians and 12.81% residents or below), 36.61% were frontline nurses, 9.28% were medical technicians and 2.87% were managers.

The respondents were predominantly woman (66.31%) and older than 45 years of age (52.43%). Nearly 52% of the respondents had worked in their hospitals for 10 years or more (online supplementary appendix A).

### Perceptions of safety climate

In this multiregion study, the mean PPR of the overall safety climate among all 54 hospitals was 9.00%. The dimensions with the four highest PPRs were 'fear of blame and punishment' (64.81%), 'fear of shame' (20.42%), 'provisions of safe care' (16.31%), and 'organisational resources for safety' (9.55%) (table 1).

| Table 1 | Problematic responses for dimensions and items |
|---|---|
| **Dimensions and items*** | **% Problematic** |
| Hospital contributions to the safety climate | |
| Senior managers' engagement | 2.84 |
| Organisational resources for safety | 9.55 |
| Overall emphasis on patient safety | 2.63 |
| Work unit contributions to the safety climate | |
| Unit managers' support | 5.42 |
| Unit safety norms | 4.51 |
| Unit recognition and support for safety efforts | 6.68 |
| Collective learning | 2.10 |
| Psychological safety | 5.68 |
| Problem responsiveness | 2.61 |
| Interpersonal contributions to the safety climate | |
| Fear of shame | 20.42 |
| Fear of blame and punishment | 64.81 |
| Other aspects of the safety climate | |
| Provision of safe care | 16.31 |
| Overall average | 9.00 |

*The means of all items in a dimension were averaged to calculate the dimension mean.

### Variations among clinical departments

After controlling for the hospital and other individual characteristics, the top four predicted PPRs for the overall safety climate were in the ED (9.63%), CADs (9.52%), paediatrics (9.46%) and the ICU (9.19%); the two lowest predicted PPRs were in the obstetrics and gynaecology department (6.82%) and anaesthesiology and the OR (7.44%).

The PPRs for 'fear of shame' and 'fear of blame and punishment' were universally very high across different types of departments. In addition, the PPRs of 'fear of shame' and 'fear of blame and punishment' in surgery departments were both the highest among the nine types of departments (23.51% and 69.93%, respectively). Moreover, the PPR of 'organisational resources for safety' in paediatrics departments was the highest across the various types of departments (14.41%) (table 2).

### Variations among job types

After adjusting for other personnel and hospital characteristics, the PPR reported by physicians was the highest for the overall safety climate (10.19%), whereas the PPR among managers was the lowest (7.45%). The PPRs for 'fear of shame' and 'fear of blame and punishment' were generally high among staff members of various job types (table 3).

Physicians perceived 'organisational resources for safety' more negatively (PPR=12.25%) than non-professionals and non-managers (PPR=6.96%). Physicians reported a higher PPR for 'unit recognition and support for safety efforts' (9.78%) than nurses (5.45%) and non-professionals and non-managers (4.48%). Physicians' perceptions of 'psychological safety' and 'problem responsiveness' were also worse than those of all other staff members (table 3, online supplementary appendix B). Additionally, the managers' responses to many dimensions seemed to be similar to those of frontline workers (online supplementary appendix C).

### Variations within working departments among physicians with various titles and other personnel

The results revealed that the PPRs reported by various types of staff members in paediatrics, ICU and CADs differed across many dimensions. For example, chief physicians and associate chief physicians in paediatrics responded more negatively than other physicians and personnel on 'organisational resources for safety' (37.03%). However, attending physicians and residents or below reported higher PPRs for 'provision of safe care' (29.56% and 26.17%, respectively) than other physicians and personnel (figure 1).

In the ICU, attending physicians seemed to respond particularly negatively across almost all dimensions, with PPRs >10%. However, chief physicians and associate chief physicians' PPR for the 'provision of safe care' was remarkably higher than that of other physicians and personnel (65.10%) (figure 1). In the CADs, the PPRs of residents or below were slightly higher across many dimensions,

**Table 2** Patient safety climate by working department*

| Dimensions | Internal medicine | | Surgery | | Obstetrics and gynaecology | |
|---|---|---|---|---|---|---|
| | Estimate | 95% CI | Estimate | 95% CI | Estimate | 95% CI |
| Hospital contributions to the safety climate | | | | | | |
| Senior managers' engagement | 2.74 | (1.58 to 3.90) | 2.06 | (0.87 to 3.25) | 1.80 | (0.21 to 3.39) |
| Organisational resources for safety | 9.38 | (6.90 to 11.86) | 7.54 | (5.00 to 10.09) | 7.02 | (3.65 to 10.40) |
| Overall emphasis on patient safety | 2.96 | (1.57 to 4.35) | 2.19 | (0.76 to 3.62) | 1.56 | (−0.38 to 3.49) |
| Work unit contributions to the safety climate | | | | | | |
| Unit managers' support | 5.21 | (3.40 to 7.01) | 4.90 | (3.05 to 6.76) | 3.15 | (0.65 to 5.64) |
| Unit safety norms | 3.85 | (2.67 to 5.03) | 3.26 | (2.05 to 4.47) | 3.92 | (2.29 to 5.55) |
| Unit recognition and support for safety efforts | 5.95 | (3.95 to 7.94) | 5.51 | (3.46 to 7.56) | 4.29 | (1.59 to 6.99) |
| Collective learning | 2.14 | (1.10 to 3.19) | 1.40 | (0.33 to 2.48) | 0.91 | (−0.53 to 2.35) |
| Psychological safety | 2.87 | (0.94 to 4.79) | 1.83 | (-0.15,3.80) | 0.72 | (−1.92 to 3.37) |
| Problem responsiveness | 0.93 | (−0.46 to 2.31) | 0.42 | (-1.00,1.84) | -0.40 | (−2.30 to 1.49) |
| Interpersonal contributions to the safety climate | | | | | | |
| Fear of shame | 19.77 | (15.84 to 23.69) | 23.51 | (19.49 to 27.53) | 12.46 | (7.26 to 17.67) |
| Fear of blame and punishment | 66.45 | (61.69 to 71.21) | 69.93 | (65.06 to 74.79) | 58.80 | (52.60 to 65.01) |
| Other aspects of the safety climate | | | | | | |
| Provision of safe care | 17.19 | (13.45 to 20.93) | 17.67 | (13.84 to 21.50) | 22.72 | (17.75 to 27.70) |
| Overall | 8.51 | (7.57 to 9.46) | 8.28 | (7.30 to 9.25) | 6.82 | (5.52 to 8.11) |
| Hospital contributions to the safety climate | | | | | | |
| Senior managers' engagement | 2.19 | (0.02 to 4.35) | 4.86 | (2.84 to 6.87) | 4.62 | (2.97 to 6.28) |
| Organisational resources for safety | 14.41 | (9.85 to 18.97) | 10.54 | (6.28 to 14.79) | 8.83 | (5.32 to 12.35) |
| Overall emphasis on patient safety | 2.60 | (−0.04 to 5.24) | 2.79 | (0.33 to 5.26) | 2.71 | (0.70 to 4.73) |
| Work unit contributions to the safety climate | | | | | | |
| Unit managers' support | 4.42 | (1.02 to 7.82) | 5.44 | (2.27 to 8.62) | 6.29 | (3.69 to 8.89) |
| Unit safety norms | 4.71 | (2.50 to 6.92) | 3.72 | (1.66 to 5.79) | 5.43 | (3.73 to 7.12) |
| Unit recognition and support for safety efforts | 8.57 | (4.93 to 12.20) | 5.10 | (1.71 to 8.48) | 8.33 | (5.52 to 11.13) |
| Collective learning | 3.17 | (1.22 to 5.13) | 2.54 | (0.71 to 4.36) | 2.60 | (1.11 to 4.10) |
| Psychological safety | 9.50 | (5.90 to 13.09) | 6.69 | (3.34 to 10.04) | 5.29 | (2.54 to 8.05) |
| Problem responsiveness | 0.75 | (−1.81 to 3.32) | 1.11 | (−1.29 to 3.50) | 2.02 | (0.05 to 4.00) |
| Interpersonal contributions to the safety climate | | | | | | |
| Fear of shame | 16.95 | (10.01 to 23.88) | 20.25 | (13.78 to 26.71) | 20.09 | (14.68 to 25.50) |
| Fear of blame and punishment | 61.47 | (53.27 to 69.67) | 60.06 | (52.40 to 67.71) | 65.68 | (59.24 to 72.13) |
| Other aspects of the safety climate | | | | | | |
| Provision of safe care | 19.48 | (12.84 to 26.12) | 22.22 | (16.03 to 28.41) | 19.94 | (14.77 to 25.10) |
| Overall | 9.46 | (7.70 to 11.21) | 9.19 | (7.56 to 10.83) | 9.63 | (8.29 to 10.98) |
| Hospital contributions to the safety climate | | | | | | |
| Senior managers' engagement | 1.87 | (−0.11 to 3.84) | 1.53 | (0.27 to 2.80) | 1.89 | (0.85 to 2.94) |
| Organisational resources for safety | 6.10 | (1.93 to 10.28) | 7.12 | (4.43 to 9.81) | 7.80 | (5.55 to 10.05) |
| Overall emphasis on patient safety | 3.26 | (0.85 to 5.68) | 2.44 | (0.93 to 3.96) | 1.87 | (0.63 to 3.12) |
| Work unit contributions to the safety climate | | | | | | |
| Unit managers' support | 2.85 | (0.26 to 5.96) | 5.80 | (3.84 to 7.77) | 4.29 | (2.67 to 5.91) |
| Unit safety norms | 2.36 | (0.34 to 4.38) | 6.12 | (4.83 to 7.40) | 3.99 | (2.93 to 5.06) |

Continued

**Table 2** Continued

| Dimensions | Internal medicine | | Surgery | | Obstetrics and gynaecology | |
|---|---|---|---|---|---|---|
| | Estimate | 95% CI | Estimate | 95% CI | Estimate | 95% CI |
| Unit recognition and support for safety efforts | 3.03 | (−0.30 to 6.36) | 8.97 | (6.80 to 11.13) | 5.34 | (3.52 to 7.15) |
| Collective learning | 0.44 | (−1.35 to 2.23) | 3.20 | (2.06 to 4.34) | 0.91 | (−0.03 to 1.85) |
| Psychological safety | 2.28 | (−1.01 to 5.56) | 8.95 | (6.86 to 11.05) | 3.05 | (1.32 to 4.78) |
| Problem responsiveness | 0.43 | (−1.92 to 2.78) | 5.50 | (4.00 to 7.01) | 0.26 | (−0.99 to 1.51) |
| Interpersonal contributions to the safety climate | | | | | | |
| Fear of shame | 21.80 | (15.44 to 28.15) | 16.22 | (11.99 to 20.45) | 15.60 | (11.99 to 19.20) |
| Fear of blame and punishment | 66.92 | (59.39 to 74.45) | 68.73 | (63.63 to 73.83) | 67.12 | (62.72 to 71.53) |
| Other aspects of the safety climate | | | | | | |
| Provision of safe care | 19.52 | (13.44 to 25.60) | 16.54 | (12.51 to 20.58) | 18.94 | (15.51 to 22.37) |
| Overall | 7.44 | (5.84 to 9.04) | 9.52 | (8.49 to 10.55) | 7.79 | (6.93 to 8.65) |

*The results reflect predicted means (estimate and 95% CI) by clinical departments based on two-level random intercept models for each dimension and the overall safety climate adjusted for other individual characteristics (age, gender, education, working years and monthly income) and hospital characteristics (tertiary level vs secondary level, hospital size, hospital location and doctor–nurse ratio). When estimating the predicted mean values of the PPR for each dimension and overall by clinical department and job type, we held other covariates constant at their means.

such as 'overall emphasis on patient safety', 'unit safety norms', 'unit recognition and support for safety efforts', 'collective learning', 'psychological safety' and 'problem responsiveness' (figure 1).

## DISCUSSION
### Patient safety climate in hospitals
Patient safety climate is a core determinant that ensures and promotes hospital safety in China. The Chinese Hospital Association has promulgated annual patient safety goals since 2007, and the National Health and Family Planning Commission (previously known as the Ministry of Health) established a web-based voluntary adverse event reporting system in 2008. Additionally, many provincial governments have actively facilitated the use of clinical pathways, disease management programmes and computer-assisted quality and safety programmes; they have also linked government subsidies for public hospitals to the quality and safety of their medical care.[37 38] These efforts all aim to facilitate an improved patient safety climate in China's hospitals.

The survey results revealed that perceptions of the overall patient climate were relatively good (PPR=9%). However, substantial attention should be paid to the dimensions 'fear of blame and punishment' (65%), 'fear of shame' (20%), 'provision of safe care' (16%) and 'organisational resources for safety' (10%) based on high-reliability organisation theories.[39] The high prevalence of 'fear of blame and punishment' and 'fear of shame' among staff members with different job types and in various working departments may be attributed to inappropriate systems for performance assessments and rewards, a hierarchical management style, difficult doctor–patient relationships

and the quintessentially Chinese notion of 'face'. Because a hierarchical management style and the quintessential notion of 'face' are common in Chinese culture, high PPRs for 'fear of blame and punishment' and 'fear of shame' may exist in other hospitals in which the dominant Chinese culture is present. However, we also found that the PPRs of these two dimensions were significantly lower than those in the previous study in Pudong New Area (78.53% and 41.16%, respectively).[22] It is possible that an increasing number of policy-makers, hospital managers and health professionals have gradually realised that concealing mistakes or errors might result in worse consequences and have therefore implemented measures such as improving reward systems, changing the patient safety climate, and establishing information systems for easier error reporting to encourage the reporting of errors or mistakes.[40 41]

### General safety climate variations by department
Our study highlighted the very high proportion of problematic responses related to 'fear of shame' and 'fear of blame and punishment' that universally appeared across the nine types of departments. This result revealed the most jarring and prevailing issues within Chinese public hospitals, as discussed above. We also found that personnel in paediatrics, the ICU, the ED and CADs perceived slightly lower levels of overall safety climate than personnel in the other working departments (PPRs were very close to 10%). Staff members in obstetrics and gynaecology departments and in anaesthesiology departments and ORs responded slightly positively for overall safety climate. For anaesthesiology and ORs, we found some previous studies with similar results, which indicates that more resources may have been provided and greater

**Table 3** Patient safety climate by job type*

| Dimensions | Physicians | | Nurses | | Medical technicians | |
|---|---|---|---|---|---|---|
| | Estimate | 95% CI | Estimate | 95% CI | Estimate | 95% CI |
| Hospital contributions to the safety climate | | | | | | |
| Senior managers' engagement | 4.01 | (3.20 to 4.81) | 2.17 | (1.28 to 3.07) | 2.76 | (0.95 to 4.58) |
| Organisational resources for safety | 12.25 | (10.48 to 14.02) | 9.97 | (8.02 to 11.92) | 8.86 | (5.04 to 12.69) |
| Overall emphasis on patient safety | 3.36 | (2.43 to 4.29) | 2.37 | (1.34 to 3.41) | 1.83 | (−0.38 to 4.04) |
| Work unit contributions to the safety climate | | | | | | |
| Unit managers' support | 6.15 | (4.92 to 7.38) | 5.50 | (4.13 to 6.86) | 4.26 | (1.41 to 7.10) |
| Unit safety norms | 4.92 | (4.11 to 5.74) | 4.99 | (4.09 to 5.90) | 2.93 | (1.07 to 4.78) |
| Unit recognition and support for safety efforts | 9.78, | (8.33 to 11.22) | 5.45 | (3.87 to 7.04) | 6.19 | (3.14 to 9.24) |
| Collective learning | 2.86 | (2.14 to 3.58) | 1.80 | (0.99 to 2.60) | 0.75 | (−0.89 to 2.39) |
| Psychological safety | 8.30 | (6.96 to 9.63) | 5.84 | (4.35 to 7.32) | 3.90 | (0.89 to 6.91) |
| Problem responsiveness | 4.09 | (3.12 to 5.07) | 1.63 | (0.56 to 2.71) | 0.10 | (−2.05 to 2.25) |
| Interpersonal contributions to the safety climate | | | | | | |
| Fear of shame | 18.39 | (15.43 to 21.35) | 19.76 | (16.56 to 22.97) | 17.48 | (11.64 to 23.33) |
| Fear of blame and punishment | 65.04 | (61.35 to 68.73) | 63.07 | (59.10 to 67.03) | 62.67 | (55.72 to 69.61) |
| Other aspects of the safety climate | | | | | | |
| Provision of safe care | 17.57 | (14.76 to 20.37) | 21.07 | (18.03 to 24.11) | 18.98 | (13.39 to 24.58) |
| Overall | 10.19 | (9.52 to 10.86) | 8.83 | (8.09 to 9.57) | 7.89 | (6.42 to 9.36) |

| Dimensions | Managers† | | Non-professionals and non-managers | |
|---|---|---|---|---|
| | Estimate | 95% CI | Estimate | 95% CI |
| Hospital contributions to the safety climate | | | | |
| Senior managers' engagement | 2.39 | (−0.06 to 4.83) | 1.76 | (0.49 to 3.03) |
| Organisational resources for safety | 5.71 | (0.56 to 10.85) | 6.96 | (4.25 to 9.67) |
| Overall emphasis on patient safety | 3.12 | (0.11 to 6.13) | 1.76 | (0.24 to 3.28) |
| Work unit contributions to the safety climate | | | | |
| Unit managers' support | 4.38 | (0.52 to 8.24) | 3.25 | (1.28 to 5.22) |
| Unit safety norms | 3.36 | (0.85 to 5.86) | 4.56 | (3.27 to 5.85) |
| Unit recognition and support for safety efforts | 4.70 | (0.60 to 8.79) | 4.48 | (2.30 to 6.65) |
| Collective learning | 1.72 | (−0.49 to 3.94) | 2.49 | (1.35 to 3.64) |
| Psychological safety | − 0.41 | (−4.48 to 3.65) | 5.26 | (3.15 to 7.36) |
| Problem responsiveness | −1.32 | (−4.22 to 1.58) | 1.61 | (0.10 to 3.12) |
| Interpersonal contributions to the safety climate | | | | |
| Fear of shame | 18.95 | (11.19 to 26.72) | 17.99 | (13.73 to 22.24) |
| Fear of blame and punishment | 66.40 | (57.25 to 75.54) | 67.92 | (62.78 to 73.06) |
| Other aspects of the safety climate | | | | |
| Provision of safe care | 20.67 | (13.24 to 28.11) | 18.50 | (14.45 to 22.56) |
| Overall | 7.45 | (5.47 to 9.43) | 8.21 | (7.18 to 9.25) |

*The results reflect predicted means (estimate and 95% CI) for clinical departments based on two-level random intercept models for each dimension and the overall safety climate adjusted for other individual characteristics (age, gender, education, working years and monthly income) and hospital characteristics (tertiary level vs secondary level, hospital size, hospital location and doctor–nurse ratio). When estimating the predicted mean values of the PPR for each dimension and overall by clinical department and job type, we held other covariates constant at their means.
†Including managers in clinical departments, CADs and administrative departments.

efforts exerted to overcome safety hazards in anaesthesiology and ORs.[15][16] Furthermore, the results showed that the respondents in surgery departments reported

the highest PPR for 'fear of shame.' In general, surgeons, who are physicians with high professional titles, have greater social identity and self-identity. However, medical

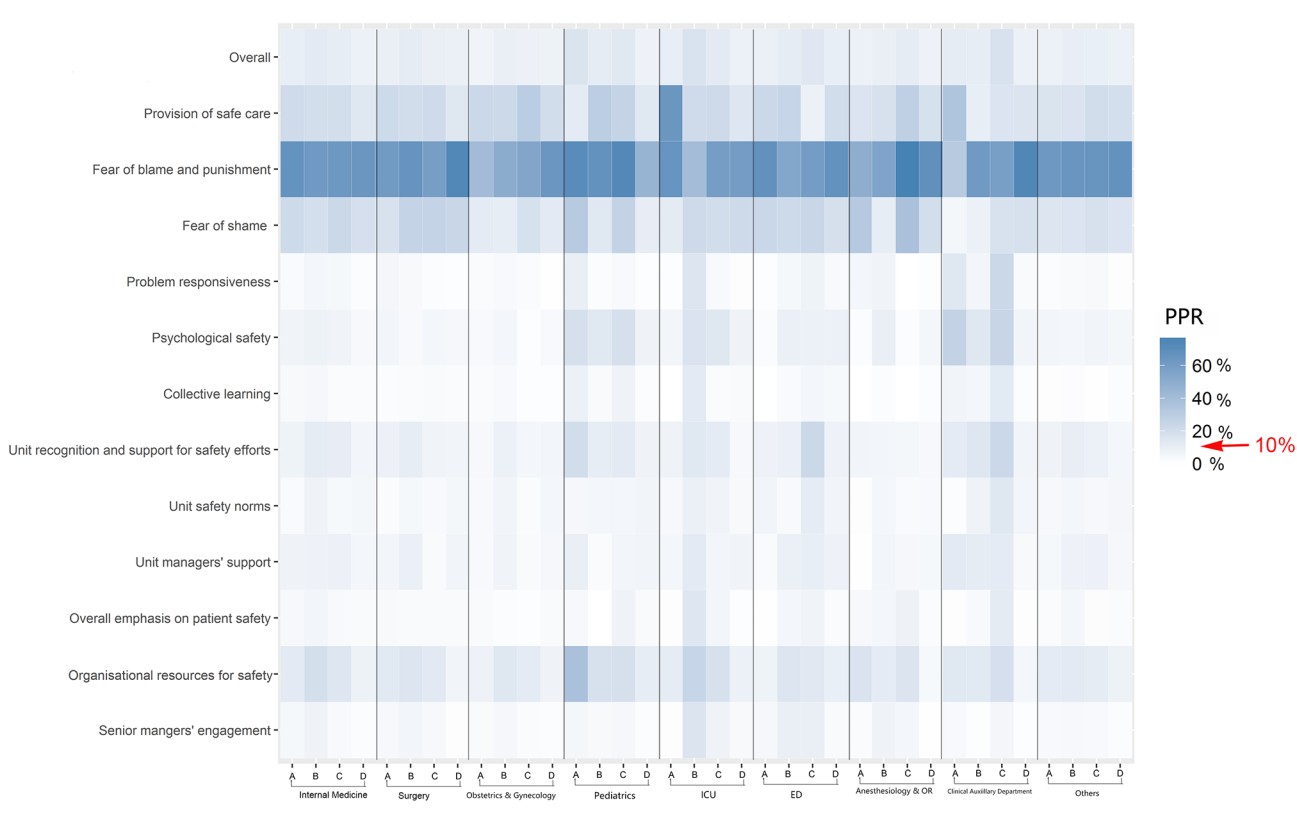

**Figure 1** Heat map of predictive means of staff among nine working departments across physicians with various titles and other staff on each dimension and the overall safety climate. ED, emergency department; ICU, intensive care unit; OR, operating room; PPR, percentages of 'problematic response'.

errors and/or accidents occurring in surgery are usually more severe than those in other working departments. Therefore, staff members in these departments experience more pressure and face more serious consequences.

### General safety climate variations by job type

The results showed that physicians' PPRs were systematically higher than those of other staff members in this study, whereas Western studies have shown the opposite.[13 16 19 42] Most frontline physicians in China are full-time employees of public hospitals. They hold dominant positions and are close to patients in the process of providing medical services. They are also responsible for ensuring patient safety and quality. If adverse events or healthcare errors occur, patients are more likely to complain and blame their physicians. Accordingly, frontline physicians might pay more attention to quality and safety measures and systems, and they also might obtain more first-hand experience or information about safety hazards.

This finding indicates that hospital managers and unit directors should establish and continuously improve internal managerial systems to encourage frontline physicians to report or share their experiences and information regarding patient safety and healthcare quality and to facilitate their proactive participation in related improvement projects. Measures to appoint frontline physicians

to serve as unit quality controllers, to encourage them to lead interdisciplinary groups to implement quality control circle activities and to involve indicators of patient safety and quality improvement in individual performance assessments have been implemented in some hospitals in China and should be generalised.[43–45]

In addition, the results also revealed that managers' perceptions were relatively consistent with those of front-line workers on many dimensions but differed from those reported in many previous studies.[22 24 34 42] This finding might be partly attributed to managers' efforts and interventions to promote 'speaking up' and 'communicating down' in China's public hospitals, including activities such as safety culture-oriented simulation training, Leadership WalkRounds and efforts to engage frontline workers and managers in open discussions about safety events.[46 47] It is also possible, however, that these results were driven by the extent to which the managers in this survey involved working department directors, in contrast to our 2013 study in Shanghai.[22] Working department directors in China are physicians who also provide healthcare for patients on the front lines.

### Paediatrics departments should receive more attention

In this investigation, paediatrics personnel responded with a high PPR (14.41%) for 'organisational resources

for safety'. In particular, chief and associate chief paediatricians were much worried about organisational resources for safety (PPR=37.03%). This is an alarming finding but is consistent with the increasingly severe shortage of paediatricians in China reported in some articles and news stories.[48–50] Statistical analyses have shown that the number of paediatricians in China increased by only 5000 from 1995 to 2010,[51] and the current number of paediatric doctors per 1000 children is only 0.5, which is one-third of the ratio in the USA. By contrast, in China, the number of obstetricians and gynaecologists per 1000 maternal age women (20–40 years of age) is estimated to be approximately 1.2, and the number of internists per 1000 adults (>14 years of age) is estimated to be nearly 0.65.[52] This issue will become more problematic following the implementation of China's two-child policy. Paediatricians continue to leave medical practice, especially in primary healthcare institutions, and many new medical graduates are unwilling to join.[53] The results also showed that attending physicians and residents with direct and close patient contact reported high PPRs for 'provision of safe care' (29.56% and 26.17%, respectively). The human resource shortage in healthcare is very likely to have a negative impact on healthcare quality and patient safety.

### ICUs and EDs should receive more attention
Working in ICUs and EDs is associated with a higher level of risk, complexity and difficulty, a faster pace and lower predictability.[15 16] These work areas are intrinsically hazardous, and personnel working in these areas are prone to have a high workload, to work long hours and to face substantial pressure, which can result in burnout.[54–57] Previous studies have indicated that burnout among healthcare providers may lead to reduced patient safety.[58–60] Nahrgang et al[61] generally argued that burned-out employees' mental and physical energy levels decrease safe work behaviours and thus increase the likelihood of errors and work-related injuries. Flinn and Armstrong conducted a post call assessment of junior doctors after extended work shifts (average 32.75 hour) and noted a significant decline in cognitive functioning and clinical decision-making performance.[62] Sharpe et al[63] showed deteriorating performance of ICU residents during 26 hours of continuous wakefulness. Because of the intrinsic characteristics of ICUs and EDs, we should attach great importance to them. In addition, the results also revealed that the predicted PPRs in ICUs and EDs for overall safety climate ranked in the top four among the working departments. It is very important and practical to design and optimise systems to protect ICU and ED patients from preventable harm. This approach requires a balanced interdisciplinary effort directed at process characteristics and the simultaneous execution of several other measures.[64] Recent examples such as standardisation of processes,[65–67] adaptation to humans' cognitive limitations,[68] optimisation of working conditions[69] and increased use of supporting information technologies should be explored.

We also found that attending ICU physicians responded more negatively than other physicians and personnel; almost all of their PPRs were >10%. These healthcare workers are important personnel on the front lines in their departments and perhaps experience more safety problems.[43] Hospital managers and policy-makers should carefully consider their opinions.

### CADs need attention
Our study showed that the perceptions of the overall patient safety climate among staff in CADs was the second worst among the nine types of departments. In particular, residents in the CADs were more worried about several dimensions, such as 'overall emphasis on patient safety', 'unit safety norms', 'unit recognition and support for safety efforts', 'collective learning', 'psychological safety' and 'problem responsiveness'. Hospital managers might not pay sufficient attention to CADs compared with other working departments, especially with respect to the residents' training and motivation in this type of department.

### Limitations
First, the data were based on self-reporting, which might involve recall/report bias. Second, because this study was cross-sectional, we could not rule out the potential for omitted variables. Although our analyses represent an important advance over prior studies because we adjusted for important known individual and hospital characteristics, unmeasured characteristics could play a role in distinguishing personnel by working department and job type. Third, the results from the 54 public general hospitals in three regions might not be generalisable to all hospitals in China, although our sample size was reasonably large and represented public hospitals in high, middle and low socioeconomic level regions and the Eastern, Central and Western regions of China.

### CONCLUSIONS
Our study highlights differences in perceptions of the patient safety climate among and within working departments and job types that have not been previously documented in China. The findings indicate that safety climate improvement efforts should involve greater attention to the climate in the ICU, the ED, paediatrics and CADs. Therefore, it would be effective and reasonable to implement specific measures to improve the patient safety climate that target specific departments and personnel.

**Acknowledgements** We gratefully acknowledge the significant contributions of the following members of the research project team: Hongbo Zhu, Xuefeng Wei and Jianjun Gu. The authors also acknowledge all hospitals that provided assistance with data collection in this research project. We truly appreciate Dr Yongfu Yu of the Department of Clinical Epidemiology at Aarhus University Hospital Olof Palmes, who provided us with important guidance on the statistical analysis.

**Contributors** DX conceived and designed the study. All authors performed the investigation, read and approved the final manuscript. PZ and ML conducted statistical analyses and interpreted the data. PZ drafted and revised the manuscript. DX revised the manuscript.

**Funding** This research project was supported by the National Natural Science Foundation of China, grant number 71473047.

**Competing interests** None declared.

**Patient consent** Detail has been removed from this case description to ensure anonymity. The editors and reviewers have seen the detailed information available and are satisfied that the information backs up the case the authors are making.

**Ethics approval** The study was approved by the Institutional Review Board of the School of Public Health, Fudan University (IRB# 2014-03-0502).

**Provenance and peer review** Not commissioned; externally peer reviewed.

**Data sharing statement** No additional data were available.

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
