## [Reviewer comments · BMJ Open]

ARTICLE DETAILS

TITLE (PROVISIONAL)	Patient Safety Climate in General Public Hospitals in China: Differences Associated with Department and Job Type Based on a Cross-sectional Survey
AUTHORS	Zhou, Ping; Bai, Fei; Tang, Huiqing; Bai, Jie; Li, Minqi; Xue, Di

VERSION 1 – REVIEW

REVIEWER	Raymond Pong Centre for Rural and Northern Health Research Laurentian University Canada
REVIEW RETURNED	16-Jan-2017

GENERAL COMMENTS	I am generally pleased with this manuscript. It addresses an important issue in China's current healthcare system. The research was based on a fairly large sample (though the hospitals chosen for the study look more like a purposive sample, rather than a random sample, but the authors have admitted that their findings may not be generalizable) and the response rate from the surveyed hospital personnel was exceptional high. The large sample size and the high response rate together lend considerable credence to the findings. The research appears to be well conducted. The instrument (PSCHO) used to measure perceived patient safety climate is appropriate, though there was no explanation why 2 questions were added to the instrument. The research methodology and statistical analysis were clearly described, and the findings well presented, though the numbers of graphs (p. 27 - p. 32) included are, I think, excessive and somewhat unwieldy. The authors may need to work with the editor to find a better way to convey the essence of the findings without overburdening the readers. However, I do have a number of concerns and suggestions for improvement. (1) The discussion of patient safety climate in hospitals in China could benefit from being embedded in a proper context. Such contextual information is particularly important for international readers who may not know much about the Chinese healthcare system. The authors could briefly describe how hospitals operate and the current situation in relation to patient safety in China. This hopefully will provide a contextual background for readers to better understand the research and appreciate its significance.
--

There may also be a need to say something about the cultural context in order for some of the findings to make sense, particularly to international readers (see my comments in the second paragraph under #3 below). The authors appeared to be aware of this as they rightly pointed out that the PSCHO contains some items that are relevant to the Chinese culture. But they did not go any further.

There is another issue concerning context. The study collected information on hospital characteristics such as hospital size (in terms of number of beds), regional locations of the hospitals, tertiary hospitals vs. secondary hospitals, etc. But such hospital characteristics were not analyzed other than being "controlled for" (p. 9, paragraph 1) in the statistical analysis. I realize that the focus of the manuscript is on differences in perception of patient safety climate between hospital departments and between hospital personnel categories, but I think it would be helpful to show how different types of hospitals differed with respect to perceived patient safety climate (at least with respect to "overall safety climate"). Again, this should provide a broader context for the analysis and for interpreting the findings.

(2) I am not at all clear about the number of hospital departments involved in the study. Supposedly, 8 departments were involved (p. 8, paragraph 3, which also mentions "and others"). But on p. 7, paragraph 4, only 5 departments were listed. And on p. 10, paragraph 1, I counted 9 departments. "OR" was mentioned in some instances, but not in others. This needs to be clarified and it needs to be consistent.

There is another issue concerning hospital departments. The study included the "medical technical department." What is medical technical department? It is not a term commonly seen in the literature. Were the authors referring to "biomedical engineering department" or "medical laboratory department"? The authors need to provide some explanation or clarification (may be in a footnote).

(3) The manuscript concluded by saying that improvements to patient safety climate needed to take different hospital departments and different hospital personnel categories into consideration, rather than using a one-size-fits-all approach. There is no disagreement to this recommendation. But, I think it would be even more helpful if the authors could offer a few concrete suggestions about how this could be done. For example, how strategies to improve patient safety climate for physicians should differ from those for hospital management? Or how strategies for ICUs should differ from those for OBGYN departments?

The very high "problematic responses" for "fear of blame and punishment" and "fear of shame" (Table 1) deserve some attention and comments in the "Discussion" section. This may have something to do with the traditional Chinese management style that emphasizes hierarchy and bureaucratic control and the quintessential Chinese notion of "face."

Given that BMJ Open has a world-wide readership, it would be helpful if the authors could briefly discuss the international implications of their study.

(4) Although I think the manuscript is well written and the standard of English is acceptable for publication, there are some minor language

	problems, which, I believe, can be fixed quite easily. If BMJ Open does not provide copy editing support, the authors need to go through the manuscript carefully and clean up the language.
--	--

REVIEWER	Liane Ginsburg York University Canada
REVIEW RETURNED	23-Jan-2017

GENERAL COMMENTS	This paper has the benefit of a large sample from a variety of regional and socioeconomic areas of China. However, I struggled with some of the data, in part because some of the methods employed did not provide sufficient detail and, in part, because the sheer volume of results were difficult to sift through. I provide more detailed comments below that elaborate on each of these two areas. In the introduction, the authors state “Because patient safety climate is associated with positive outcomes, such as greater error reporting,⁵ less adverse events,^{6,7} lower mortality rates,⁸”. Reference 8 to Estabrooks et al 2002 paper is not about PS climate or mortality. In the sample section on page 6 it would be useful for international readers not familiar with the geographic jurisdictions (eg what is the overlap between prefectures, vs districts vs cities, etc.) used in China to have this explained with greater clarity Under data sources (page 7) it is not clear if the number of people you sampled in each group on the employee survey is per hospital or per department. In a smaller setting where you sampled 15 physicians is this across an entire hospital or 15 physicians per department? It would also be useful to provide some further information on sampling: how you identified staff – did you have access to hospital staff lists? How accurate are these? Did you sample full-time employees only? What was the total number of surveys that were sent out? Data on survey procedures is also needed: How were surveys distributed and collected (electronic? Paper?, was their follow up for non-responders?) I didn't see and ethics statement Page 9. There does not seem to be any data on the survey response rate. This needs to be provided. Without the kinds of information on survey procedures noted in my previous point it is not possible to know much about the representativeness of your respondent group Page 10 – respondent groups as a proportion of total respondents reported (physicians, nurses etc) on page 10 adds to 90% who are the other 10%? Pp 10-11, for results in Table 2 – are the significant differences noted by the p values for that department compared to all other departments or only compared to some depts. – were pairwise comparisons made?
--

	All of the graphs in Figure 1 were overwhelming – lots of data and it's difficult to get a sense of the magnitude of the differences that are being shown. There are also no Ns on these figures and it seems like there would be a relatively small numbers for some of the subgroup presented (Eg assocatie physicians in ICU or anesthesia). You present a tremendous amount of data and the discussion should be used to highlight the most salient aspects of your data and to relate your findings back to the large volume of literature that exists on safety climate perceptions in different settings / among different work groups. After reading the discussion I feel unconvinced of the key messages from this work beyond the fact that fear of blame and shame continue to be tremendously problematic aspects of pf safety climate. As an aside, your work might also be strengthened by examing the strength of climate perceptions – it is hard to tell but you may have a sufficient sample size to look at the extent to which staff in different departments “agree” about their safety climate ratings (see Ginsburg & Oore, 2016, BMJQS for a discussion of looking at safety climate strength in addition to climate level which you focus on)
--	--

REVIEWER	Ricardo Segurado University College Dublin, Ireland
REVIEW RETURNED	07-Feb-2017

GENERAL COMMENTS	Thank-you for the opportunity to review. This is a large, important, and well-conducted study. It acheived an impressive response rate, and the design and sampling of the study is well described, and pragmatic. I have some minor requests which should be easy to correct, and some optional comments. I have one major revision request which I hope the authors and editors agree with. Minor comments:  - I ask the authors to mention where and when ethical approval was obtained for the study, and how participants' consent was obtained. - The authors should amend the Methods - Data Sources section (page 7) to clarify whether Pediatrics, Med Tech, Anaesthesiology employees were recruited, in addition to the ones mentioned, and whether the groups were combined (e.g. Anaesthes. and OR). - In Methods - Survey Data Analysis - Psychometric Analysis, the authors discuss the results of the confirmatory factor analysis on the PSCHO tool, which appear satisfactory. One of the references they specify advises thresholds around >0.95 for fit statistics such as the CFI, NFI: I request the authors give the actual obtained values rather than stating they were >0.9. This would make it consistent with the RMSEA, sRMR and GFI statistics where the actual estimates are presented. - Statistical Analysis methods should specify how any categorical predictors (sex, education, etc) were handled in the calculation of "predicted" means, alongside the mention that other covariates were "held constant at their means". The same comment applies to the legend of the table in Appendix B.
---

- On page 10, "Appendix 1" should be replaced with "Appendix A"

- Figure 1 might be revised in the context of my comments on statistical significance, below; it is quite large and the authors might best focus on up to 4, 6 or maybe 8 plots, with all plots presented in supplemental materials / appendix - for discussion with editors.

- (optional) The psychometric performance of the PSCHO was good in my judgement, but on the borderline of what is considered "good"; if the authors conducted any sensitivity analysis on their two additional items, it would be worth reporting this

- (optional) The authors validate the PSCHO tool in their sample using CFA and report (in the methods section) the results of this, however their analysis focused on a different construct, derived from the PSCHO: the PPR. I would strongly urge the authors to explore the validity of the PPR construct in future work, as it may be quite different from the PSCHO. I also note that the PPR used in this paper is defined differently than in their previous publication (ref 22), which used <4 or >4 as thresholds for problematic response.

- (optional) I suggest that Figure 1 plots would be revised to present a symbol at each mean, with appropriate error bars, instead of line-plots which are usually used for longitudinal change

Major comment:

- I believe the p-values or "statistical significance" indicators in Tables 2 and 3 arise from post-hoc comparisons performed in the hierarchical linear models. If so, these compare each categorical predictor groups. I strongly urge the authors to remove p-value / "significance" information from Tables 2 and 3. In exploratory modelling such as this, the role and validity of p-values is highly questionable, and Table 2 and 3 do not benefit from a focus on "statistical significance". 95% confidence intervals should be given for the predicted means.

To convince the authors of this, I note that 1) p-values should be corrected for multiple testing if they are to control type I error; 2) the type II error rate for each test producing a p-value will be different - perfectly illustrated in several places in Table 2, where a large PPR is not significant (e.g. Collective learning in ICU) whereas a smaller PPR is significant (e.g. Collective learning in Internal Medicine); 3) Comparisons between each level and all other levels (Table 2), between each level and a reference level (Managers, in Table 3), or between each level and a different reference level, will all produce different p-values, and it is not clear which of these options is more valid than any other.

In summary, the purpose of this modelling is to identify problem dimensions, within subgroups of job title and hospital department. To achieve this aim the authors should assess the absolute values of the predicted means, and some measure of uncertainty such as a standard error or a confidence interval.

The Results and Discussion sections, which focus on "significant" results, must be updated. The authors could decide on a threshold to decide on whether a mean is low enough or high enough to be

	discussed further (e.g. >10% already emerges on page 12 as an ad hoc threshold). I suggest that any comparisons between job titles or between departments is conducted subjectively and with an eye on the confidence intervals. Visualisation tools (e.g. a heat map) might be usefully applied here. And finally, the Table in Appendix B should also drop the statistical significance indicators. For an overview of such a large number of models and predictor levels, again a heat-map based on the means would be preferred.
--	---

REVIEWER	Stephen Senn Luxembourg Institute of Health, Luxembourg
REVIEW RETURNED	10-Feb-2017

GENERAL COMMENTS	A general comment is that the statistical analysis seems plausibly appropriate but given the complexity of the study and the limitations on space it is difficult to be certain that it is OK. Further specific comments follow. The order of the comments is order in the manuscript and not order of importance. (So trivial and more important comments are mixed.) 1) Abstract. It is not clear to me whether your analysis truly is multivariate (multiple LH outcome terms simultaneously studied) or merely multivariable (multiple RH predictor terms). I suspect the latter. See https://www.ncbi.nlm.nih.gov/pmc/articles/PMC3518362/ and please clarify. 2) P4 L15 'fewer adverse events' 3) P5 L51. Although 'fear of blame' may be important in Chinese culture, the relevance of this to safety is not immediately obvious. For example, it might suppress willingness to report problems but on the other hand it might increase desire to avoid them. It is not clear here as to how fear of blame aspect is incorporated in the measuring instrument. A brief explanation might be helpful. 4) P7 L49 etc It is not clear how validity etc were tested. 5)p8 L58 It is not clear what is meant by 'hospital membership' 6) P9 L53. Please state how many staff were invited to respond. 7) P11 L28. It is not clear what 'personnel' characteristics are. If you were comparing different personnel you would not adjust for them. Do you mean demographic characteristics? 8) Table 2. It is not at all clear what you are doing here as regards your P-values. Is this a comparison of each department to all others or only to those labelled as 'others'? It is not clear what this should be of interest. The department effect overall (as a variance) might be more relevant. it was hard to follow what you were doing her.
---

VERSION 1 – AUTHOR RESPONSE

Dear Dr. Adrian Aldcroft and Reviewers,

Thank you for your letter and for the reviewers' comments concerning our manuscript entitled "Patient Safety Climate in General Public Hospitals in China: Multilevel Modeling of Differences Associated with Department and Job Type". Those comments are all valuable and very helpful for revising and improving our paper. We have read comments carefully and have made revision which we hope meet with approval. Revised portion are highlighted in yellow and/or with tracked changes in the paper. Our point-by-point responses are listed below.

Responds to the reviewers' comments:

Reviewer: 1

Reviewer Name: Raymond Pong

Institution and Country : Centre for Rural and Northern Health Research, Laurentian University, Canada

Please state any competing interests or state 'None declared': None declared

Please leave your comments for the authors below

I am generally pleased with this manuscript. It addresses an important issue in China's current healthcare system. The research was based on a fairly large sample (though the hospitals chosen for the study look more like a purposive sample, rather than a random sample, but the authors have admitted that their findings may not be generalizable) and the response rate from the surveyed hospital personnel was exceptional high. The large sample size and the high response rate together lend considerable credence to the findings.

The research appears to be well conducted. The instrument (PSCHO) used to measure perceived patient safety climate is appropriate, though there was no explanation why 2 questions were added to the instrument. The research methodology and statistical analysis were clearly described, and the findings well presented, though the numbers of graphs (p. 27 - p. 32) included are, I think, excessive and somewhat unwieldy. The authors may need to work with the editor to find a better way to convey the essence of the findings without overburdening the readers.

Response: We have added the reasons why 2 questions had been added to the instrument (P. 6, L9-11). We have also revised our graphs to make it more clear.

However, I do have a number of concerns and suggestions for improvement.

(1) The discussion of patient safety climate in hospitals in China could benefit from being embedded in a proper context. Such contextual information is particularly important for international readers who may not know much about the Chinese healthcare system. The authors could briefly describe how hospitals operate and the current situation in relation to patient safety in China. This hopefully will provide a contextual background for readers to better understand the research and appreciate its significance.

Response: We have added contextual information about patient safety in China in the first paragraph of the discussion, named "Patient Safety Climate in Hospitals" (P12, L19-29).

There may also be a need to say something about the cultural context in order for some of the findings to make sense, particularly to international readers (see my comments in the second paragraph under #3 below). The authors appeared to be aware of this as they rightly pointed out that the PSCHO contains some items that are relevant to the Chinese culture. But they did not go any further.

Response: According to the suggestion, we have added statements to make it clearer (P5, L23-28).

There is another issue concerning context. The study collected information on hospital characteristics such as hospital size (in terms of number of beds), regional locations of the hospitals, tertiary hospitals vs. secondary hospitals, etc. But such hospital characteristics were not analyzed other than being "controlled for" (p. 9, paragraph 1) in the statistical analysis. I realize that the focus of the manuscript is on differences in perception of patient safety climate between hospital departments and between hospital personnel categories, but I think it would be helpful to show how different types of hospitals differed with respect to perceived patient safety climate (at least with respect to "overall

safety climate"). Again, this should provide a broader context for the analysis and for interpreting the findings.

Response: Because this paper was aimed to analyze the variation of perception on patient safety climate among different departments and among different job types, the hospital characteristics were used as control variables. In fact, we have analyzed the differences in perceived patient safety climate among different types of hospitals in another paper. Therefore, we did not make changes for this comment.

(2) I am not at all clear about the number of hospital departments involved in the study. Supposedly, 8 departments were involved (p. 8, paragraph 3, which also mentions "and others"). But on p. 7, paragraph 4, only 5 departments were listed. And on p. 10, paragraph 1, I counted 9 departments. "OR" was mentioned in some instances, but not in others. This needs to be clarified and it needs to be consistent.

Response: Nine types of departments were involved in this study. Therefore, we have corrected the related descriptions in the paper (P7, L28-30; P9, L10-13).

There is another issue concerning hospital departments. The study included the "medical technical department." What is medical technical department? It is not a term commonly seen in the literature. Were the authors referring to "biomedical engineering department" or "medical laboratory department"? The authors need to provide some explanation or clarification (may be in a footnote).

Response: We have changed the words of "medical technical departments" to "clinical auxiliary departments" in the paper. We have added the statement, "The clinical auxiliary departments are referred to laboratory department, image department, ECG department, pathology department, pharmacy department, supply center, etc." in the section of "data sources" (P7, L30-P8 L2).

(3) The manuscript concluded by saying that improvements to patient safety climate needed to take different hospital departments and different hospital personnel categories into consideration, rather than using a one-size-fits-all approach. There is no disagreement to this recommendation. But, I think it would be even more helpful if the authors could offer a few concrete suggestions about how this could be done. For example, how strategies to improve patient safety climate for physicians should differ from those for hospital management? Or how strategies for ICUs should differ from those for OBGYN departments?

Response: We have revised the discussion and added the recommendations for strengthening patient safety climate in different types of departments (P14 L12-20; P14 L24-28; P16 L5-12).

The very high "problematic responses" for "fear of blame and punishment" and "fear of shame" (Table 1) deserve some attention and comments in the "Discussion" section. This may have something to do with the traditional Chinese management style that emphasizes hierarchy and bureaucratic control and the quintessential Chinese notion of "face."

Response: According to reviewer's suggestion, we have added some explanations for high problematic responses for "fear of blame and punishment" and "fear of shame" in the first paragraph of the discussion (P13, L5-11)..

Given that BMJ Open has a world-wide readership, it would be helpful if the authors could briefly discuss the international implications of their study.

Response: We have added a statement to indicate the international implications of our study (P14 L12-20; P14 L24-28; P16 L5-12).

(4) Although I think the manuscript is well written and the standard of English is acceptable for publication, there are some minor language problems, which, I believe, can be fixed quite easily. If BMJ Open does not provide copy editing support, the authors need to go through the manuscript carefully and clean up the language.

Response: According to the suggestion, we have referred to the American Journal Experts Corporation to edit the manuscript.

Reviewer 2:

Reviewer Name: Liane Ginsburg

Institution and Country: York University, Canada

Please state any competing interests or state 'None declared': None declared

Please leave your comments for the authors below

This paper has the benefit of a large sample from a variety of regional and socioeconomic areas of China. However, I struggled with some of the data, in part because some of the methods employed did not provide sufficient detail and, in part, because the sheer volume of results were difficult to sift through. I provide more detailed comments below that elaborate on each of these two areas.

In the introduction, the authors state "Because patient safety climate is associated with positive outcomes, such as greater error reporting,⁵ less adverse events,^{6,7} lower mortality rates,⁸".

Reference 8 to Estabrooks et al 2002 paper is not about PS climate or mortality.

Response: Sorry for our mistake. We have revised the references.

In the sample section on page 6 it would be useful for international readers not familiar with the geographic jurisdictions (eg what is the overlap between prefectures, vs districts vs cities, etc.) used in China to have this explained with greater clarity

Response: According to the suggestion, we have added related information to clarify it in the section of "Sample" (P6 L17,20,23-25; P7 L1).

Under data sources (page 7) it is not clear if the number of people you sampled in each group on the employee survey is per hospital or per department. In a smaller setting where you sampled 15 physicians is this across an entire hospital or 15 physicians per department?

Response: The number of people we sampled in each group was calculated at the hospital level. In a smaller setting, we sampled 15 physicians across an entire hospital. We revised the statement to make it clearer (P7, L23).

It would also be useful to provide some further information on sampling: how you identified staff – did you have access to hospital staff lists? How accurate are these? Did you sample full-time employees only? What was the total number of surveys that were sent out?

Data on survey procedures is also needed: How were surveys distributed and collected (electronic? Paper?, was their follow up for non-responders?)

Response: We only sampled full-time employees and sent out 4753 questionnaires totally. The survey of hospital employees was conducted using anonymous, paper-based, self-administered questionnaires. The questionnaires were distributed and collected by trained coordinators in the surveyed hospitals or regions according to our study design. However, we did not follow up for non-responders, because the survey was anonymously conducted.. We have added more relevant information in this section (P7, L16-18).

I didn't see an ethics statement

Response: We have added this information (P8, L3-4).

Page 9. There does not seem to be any data on the survey response rate. This needs to be provided. Without the kinds of information on survey procedures noted in my previous point it is not possible to know much about the representativeness of your respondent group

Response: We have added related information (P10, L17-18).

Page 10 – respondent groups as a proportion of total respondents reported (physicians, nurses etc) on page 10 adds to 90% who are the other 10%?

Response: The others were non-professionals and non-managers. We have added this information in “respondent characteristics” in the section of results and in the note of Appendix A.

Pp 10-11, for results in Table 2 – are the significant differences noted by the p values for that department compared to all other departments or only compared to some depts. – were pairwise comparisons made?

Response: In the original Table 2, the significant differences were noted by the p values for that department compared to "others" departments not to all the other departments. But in the revised manuscript, we used a heatmap (Figure 1) to replace the original Table 2, according to a reviewer's suggestion.

All of the graphs in Figure 1 were overwhelming – lots of data and it's difficult to get a sense of the magnitude of the differences that are being shown. There are also no Ns on these figures and it seems like there would be a relatively small numbers for some of the subgroup presented (Eg associate physicians in ICU or anesthesia).

Response: We have changed this figure to a heatmap (Figure 1), as stated above.

You present a tremendous amount of data and the discussion should be used to highlight the most salient aspects of your data and to relate your findings back to the large volume of literature that exists on safety climate perceptions in different settings / among different work groups. After reading the discussion I feel unconvinced of the key messages from this work beyond the fact that fear of blame and shame continue to be tremendously problematic aspects of patient safety climate.

Response: We have revised our descriptions in the section of results to focus on the differences of perceived patient safety climate among different working departments and job types and on key messages from our study.

As an aside, your work might also be strengthened by examining the strength of climate perceptions – it is hard to tell but you may have a sufficient sample size to look at the extent to which staff in different departments “agree” about their safety climate ratings (see Ginsburg & Oore, 2016, BMJQS for a discussion of looking at safety climate strength in addition to climate level which you focus on)

Response: According to the suggestion, we have analyzed the variations of the predicted means of perception of patient safety climate on each dimension and overall among physicians with different titles and other personnel within each of 9 departments.

Reviewer: 3

Reviewer Name: Ricardo Segurado

Institution and Country : University College Dublin, Ireland

Please state any competing interests or state ‘None declared’: None declared

Please leave your comments for the authors below

Thank you for the opportunity to review. This is a large, important, and well-conducted study. It achieved an impressive response rate, and the design and sampling of the study is well described,

and pragmatic. I have some minor requests which should be easy to correct, and some optional comments. I have one major revision request which I hope the authors and editors agree with.

Minor comments:

- I ask the authors to mention where and when ethical approval was obtained for the study, and how participants' consent was obtained.

Response: This study was approved by the Institutional Review Board of the School of Public Health, Fudan University (IRB# 2014-03-0502) and written informed consent was exempted because our survey of employees was anonymous and had risks less than minimum. During the survey, anonymous, self-administered questionnaires were distributed and collected by trained coordinators in the surveyed hospitals or regions according to our study design. If the sampled employees were willing to participate in our study, they filled the questionnaires and returned them to the coordinators. (P7 L16-18)

We added the related information in the "Data Source".

- The authors should amend the Methods - Data Sources section (page 7) to clarify whether Pediatrics, Med Tech, Anaesthesiology employees were recruited, in addition to the ones mentioned, and whether the groups were combined (e.g. Anaesthes. and OR).

Response: We have modified the relevant descriptions in "Data Source" to clarify the types of departments in this study (P7,L28-30; P9 L10-13).

- In Methods - Survey Data Analysis - Psychometric Analysis, the authors discuss the results of the confirmatory factor analysis on the PSCHO tool, which appear satisfactory. One of the references they specify advises thresholds around >0.95 for fit statistics such as the CFI, NFI: I request the authors give the actual obtained values rather than stating they were >0.9 . This would make it consistent with the RMSEA, sRMR and GFI statistics where the actual estimates are presented.

Response: We have added the actual values of CFI, NFI and NNFI (P8, L28-P9,L2) and revised the references according to the suggestion.

- Statistical Analysis methods should specify how any categorical predictors (sex, education, etc) were handled in the calculation of "predicted" means, alongside the mention that other covariates were "held constant at their means". The same comment applies to the legend of the table in Appendix B.

Response: We have revised the related statement and have added these information in some related tables (P9 L21-23)

- On page 10, "Appendix 1" should be replaced with "Appendix A"

Response: We have corrected it.

- Figure 1 might be revised in the context of my comments on statistical significance, below; it is quite large and the authors might best focus on up to 4, 6 or maybe 8 plots, with all plots presented in supplemental materials / appendix - for discussion with editors.

Response: We have revised the original Figure 1 according to your suggestion.

- (optional) The psychometric performance of the PSCHO was good in my judgement, but on the borderline of what is considered "good"; if the authors conducted any sensitivity analysis on their two additional items, it would be worth reporting this.

Response: Based on the suggestion, we have compared the psychometric performance between the original PSCHO (without two additional items) and the PSCHO in this study. The result was as below (see Table 1.). The fit indexes of the two scales were similar.

Table 1. The psychometric performance of the PSCHO in this study and the original PSCHO
 指标 PSCHO

	(in this study)	Original PSCHO (without two additional items)	Criteria
RMR	0.0407	0.0415	<0.08
SRMR	0.0487	0.0489	<0.08
GFI	0.8436	0.8465	>0.90
AGFI	0.8260	0.8283	>0.90
PGFI	0.7937	0.7934	>0.50
RMSEA	0.0580	0.0589	<0.08
CFI	0.9125	0.9130	>0.90
NFI	0.9060	0.9067	>0.90
NNFI	0.9070	0.9072	>0.90

- (optional) The authors validated the PSCHO tool in their sample using CFA and report (in the methods section) the results of this, however their analysis focused on a different construct, derived from the PSCHO: the PPR. I would strongly urge the authors to explore the validity of the PPR construct in future work, as it may be quite different from the PSCHO. I also note that the PPR used in this paper is defined differently than in their previous publication (ref 22), which used <4 or >4 as thresholds for problematic response.

Response: We will try to explore the validity of the PPR construct in future work. And in the previous study, we used a 7-point Likert scale. In this time, we used a 5-point Likert scale according to the original PSCHO. So the PPR used in this paper is defined differently than in the previous publication.

- (optional) I suggest that Figure 1 plots would be revised to present a symbol at each mean, with appropriate error bars, instead of line-plots which are usually used for longitudinal change

Response: According to the suggestions from the reviewers, we have used a heatmap to replace the original Figure 1.

Major comment:

- I believe the p-values or "statistical significance" indicators in Tables 2 and 3 arise from post-hoc comparisons performed in the hierarchical linear models. If so, these compare each categorical predictor groups. I strongly urge the authors to remove p-value / "significance" information from Tables 2 and 3. In exploratory modelling such as this, the role and validity of p-values is highly questionable, and Table 2 and 3 do not benefit from a focus on "statistical significance". 95% confidence intervals should be given for the predicted means.

Response: According to the suggestion, we have revised the original Tables 2 and 3. Both of the predicted means and their 95% confidence intervals were reported and the p-values/"statistical significance" indicators were removed. The revised tables were presented in the supplemental material (Appendix B and Appendix C). And we used heatmaps to display the results in the section of Results.

To convince the authors of this, I note that 1) p-values should be corrected for multiple testing if they are to control type I error; 2) the type II error rate for each test producing a p-value will be different - perfectly illustrated in several places in Table 2, where a large PPR is not significant (e.g. Collective learning in ICU) whereas a smaller PPR is significant (e.g. Collective learning in Internal Medicine); 3) Comparisons between each level and all other levels (Table 2), between each level and a reference level (Managers, in Table 3), or between each level and a different reference level, will all produce different p-values, and it is not clear which of these options is more valid than any other.

Response: Thanks a lot for the detailed explanations.

In summary, the purpose of this modelling is to identify problem dimensions, within subgroups of job title and hospital department. To achieve this aim the authors should assess the absolute values of the predicted means, and some measure of uncertainty such as a standard error or a confidence interval.

Response: We have modified the relevant statement according to the suggestion in appendix and context.

The Results and Discussion sections, which focus on "significant" results, must be updated. The authors could decide on a threshold to decide on whether a mean is low enough or high enough to be discussed further (e.g. >10% already emerges on page 12 as an ad hoc threshold). I suggest that any comparisons between job titles or between departments is conducted subjectively and with an eye on the confidence intervals. Visualisation tools (e.g. a heat map) might be usefully applied here.

Response : Thanks a lot for the suggestions. We have used heatmaps in the revised paper and focused our descriptions and discussion on what you suggested.

And finally, the Table in Appendix B should also drop the statistical significance indicators. For an overview of such a large number of models and predictor levels, again a heat-map based on the means would be preferred.

Response: According to the suggestion, we have used heat maps based on the predicted means and dropped the statistical significance indicators in the original Appendix B.

Reviewer: 4

Reviewer Name : Stephen Senn

Institution and Country : Luxembourg Institute of Health, Luxembourg

Please state any competing interests or state 'None declared':

None as far as I am aware. My general online declaration of interest is here
http://www.senns.demon.co.uk/Declaration_Interest.htm

Please leave your comments for the authors below

A general comment is that the statistical analysis seems plausibly appropriate but given the complexity of the study and the limitations on space it is difficult to be certain that it is OK. Further specific comments follow. The order of the comments is order in the manuscript and not order of importance. (So trivial and more important comments are mixed.)

1) Abstract. It is not clear to me whether your analysis truly is multivariate (multiple LH outcome terms simultaneously studied) or merely multivariable (multiple RH predictor terms). I suspect the latter. See <https://www.ncbi.nlm.nih.gov/pmc/articles/PMC3518362/> and please clarify.

Response: We have corrected it (P2, L12).

2) P4 L15 'fewer adverse events'

Response: We have corrected it (P4, L7).

3) P5 L51. Although 'fear of blame' may be important in Chinese culture, the relevance of this to safety is not immediately obvious. For example, it might suppress willingness to report problems but on the other hand it might increase desire to avoid them. It is not clear here as to how fear of blame aspect is incorporated in the measuring instrument. A brief explanation might be helpful.

Response: Based on the suggestion, we have added the explanation in the context (P5, L23-28).

4) P7 L49 etc It is not clear how validity etc were tested.

Response: We have added the statement (P8, L9-11).

5) p8 L58 It is not clear what is meant by 'hospital membership'

Response: We have corrected it (P9, L25-27-P10 L2).

6) P9 L53. Please state how many staff were invited to respond.

Response: We sent out 4753 questionnaires totally. And the response rate was added in the section of Results (P10, L18)

7) P11 L28. It is not clear what 'personnel' characteristics are. If you were comparing different personnel you would not adjust for them. Do you mean demographic characteristics?

Response: According to the suggestion, we have added the list of personnel characteristics (gender, age, education, length of working years, and monthly income) that were used in the study to control other individual covariates in the related sections.

8) Table 2. It is not at all clear what you are doing here as regards your P-values. Is this a comparison of each department to all others or only to those labelled as 'others'? It is not clear what this should be of interest. The department effect overall (as a variance) might be more relevant. it was hard to follow what you were doing her.

Response: According to the comments from the reviewers, we have revised the original Table 2 and Table 3. Both of the absolute values and 95% confidence intervals for the predicted means were reported and the p-values/"statistical significance" indicators were dropped. The revised tables were presented in the supplemental material (Appendix B and Appendix C). And we used heatmap figures to display the results.

VERSION 2 – REVIEW

REVIEWER	Liane Ginsburg York University Canada
REVIEW RETURNED	09-May-2017

GENERAL COMMENTS	The authors have attended to several concerns raised in the previous review. The strength of this paper is the extensive Chinese sample allowing the authors to contribute to an understanding of PSC perceptions in that jurisdiction. Below I note some concerns re the results – mainly it seems like a number of small, non-significant differences between groups (eg depts., staff groups) are highlighted in the results and discussion and this has the effect of playing down some of the more interesting differences that were found. Concrete instances of this concern are elaborated on below. IT may just be that the results need to be described more clearly and succinctly – it is difficult to tell. More specific things that need to be addressed (in the order they appear in the manuscript) are as follows: Abstract – In 19 – not sure what is meant by “should obtain more concerns” Several typos need to be corrected: - Abstract results section- Intoduction section, In 8 readmission rates should be followed by a comma not a period
--

- page 5. Ln 18 should say “whether THERE WERE differences by job type...”
- Methods section p5 ln 26 – say “includes “fear of blame...” instead of USE “fear of blame
- Data sources section, ln 30 – clarify what et al departments include. And also clarify if you included pediatrics and other less critical departments – these are not listed here on pp7-8 yet you mention in the study introduction that these departments are less studied (and you mention them further on in the paper) – pls make sure pp7-8 description of units in the sample is accurate.
- reference 32 is not included on your reference list
- clarity of written English needs to be checked on additions that were made to this version of the manuscript

Page 10 – Analysis section – please describe what a heat map is. Also these were included in the HTML version of the uploaded manuscript but they are not referred to or explained at all in the results.

Variation among working departments and job types (pp11-12) : it seems to make sense to only highlight differences for departments of job types where the Cis do not overlap. You mention many differences in the text but in the table the groups have overlapping Cis.

Page 11 (Variations within Working Departments among Physicians with Various Titles and Other Personnel) – there is very little text here describing a huge number of results presented in a table. The text needs to highlight and describe any statistically and clinically significant differences. Also the text refers readers to figure 3 but the supplementary file with tables has each labeled as Appendices not figures – this needs to be made consistent

Your results on the very high proportion of problematic responses related to “fear of blame and punishment” and “fear of shame” seem to be the most jarring and unique – it may help to focus on these results and the implications for safety and safety improvement as this is in many ways a dimension of PSC that reflects the very essence of the construct.

The Discussion on page 13 of climate variations by dept does not seem to fit the results in Appendix B given the overlapping Cis so I am not clear why these “differences” are highlighted in the discussion. Similarly, you state in the discussion that PPRs by physicians were systematically higher than other staff in the study but my read of Appendix C is that the docs, nurses and techs (all the clinical staff) have overlapping Cis and that the clinical groups scores show more PPRs than the managers and other non-clinicians. Unless I am reading these data incorrectly, these results are similar to what we find in other countries (managers PSC scores are nearly always more positive as many PSC items are about managerial behaviors and priorities!)

The data don’t seem to support the “Substantial Concerns in the Pediatric Departments” – their results in Appendix seem similar to other depts, except maybe OB which is lower. Is the Appendix E findings this piece of discussion is based on?

The strength of this study is the extensive nature of the sample. Accordingly, I want to see some more discussion of

	representativeness in the discussion section – of hospital type (eg secondary), hospital size (eg large), of proportion of staff in each clinical dept surveyed and of the job types. Please present some data to show whether the proportions in your respondent group (described on page 10 in the 1st results paragraph) are representative of the proportions in China? Lastly, you state in the response to reviewers that you attended to our suggestion to look at the extent to which staff in different departments “agree” about their safety climate ratings (see Ginsburg & Oore, 2016, BMJQS for a discussion of looking at safety climate strength in addition to climate level which you focus on) . You responded that “we have analyzed the variations of the predicted means of perception of patient safety climate on each dimension and overall among physicians with different titles and other personnel within each of 9 departments.” – this is not what climate strength is about and while you certainly do not NEED to performance these analyses – I want to make sure that your rationale for providing variations in predicted means is clear and what you intended to do.
--	---

REVIEWER	Raymond Pong Centre for Rural and Northern Health Research Laurentian University Canada
REVIEW RETURNED	13-May-2017

GENERAL COMMENTS	Having read the revised manuscript, I believe the authors have addressed most of the concerns I raised in my earlier review. I will let the other reviewers decide whether revisions made have adequately addressed their concerns. However, I should point out that there are still some language deficiencies (e.g., grammatical errors and problematic expressions) in the manuscript. The authors mentioned that they had used the services of a commercial editing agency to help improve the writing. But, the agency did NOT do a very good job. Here are a few examples of problematic writing: * "We specifically explore... (3) whether differences by job type within the selected clinical departments." [suggested changes: "We specifically explore... (3) whether there were differences by job type in the selected clinical departments."] * "The clinical auxiliary departments are referred to laboratory department, image department..." [suggested changes: "'Clinical auxiliary departments' refers to laboratory department, imaging department..."] * "In this survey, the result revealed that the overall perception of patient climate was relatively good (PPR=9%)." [suggested changes: "The survey results revealed that the perception of overall safety climate was relatively good (PPR=9%)."] * "Besides, the results revealed that the perceptions by managers were relatively consistent with frontline workers on many dimensions, which was different with many previous researches." [suggested changes: "Besides, the results revealed that the
--

	perceptions of managers were relatively consistent with those of frontline workers on many dimensions, but were different than those reported in many previous studies."] There are other similar language problems. I would leave it to the editor to determine whether such minor linguistic blemishes are serious enough to tarnish an otherwise substantial, well-conducted, and worthwhile study and whether some additional effort to clean up the language is needed. I also would like to make another language-related suggestion. When referring to departments within a hospital, the authors used various terms: "working departments," "work departments," and "clinical departments." I think some consistency in terminology would be helpful and suggest they use the term "clinical departments" throughout the manuscript.
--	---

VERSION 2 – AUTHOR RESPONSE

Editor, Journal of BMJ Open
Sep 8th, 2017

Dear Dr. Emma Gray and Reviewers:

We greatly appreciate the significant attention of the editor and reviewers to the manuscript "Patient Safety Climate in General Public Hospitals in China: Multilevel Modeling of Differences Associated with Department and Job Type" (Manuscript ID bmjopen-2016-015604.R1). We are particularly grateful for your comments on our paper, and we have further revised the manuscript based on those comments. Our point-by-point responses follow. In the manuscript, we have highlighted our changes in yellow.

Sincerely,

Ping Zhou

Responses to the editor's comments:

1. Please edit your title: please insert the study design into the second half of the title and remove 'multilevel modeling'.

Based on this comment, we have changed the title to "Patient Safety Climate in General Public Hospitals in China: Differences Associated with Department and Job Type based on a Cross-sectional Survey."

2. You responded to a reviewer comment about consent - please also include this information in your methods.

We have added this information (P8, lines 6-11).

3. Please ensure that your manuscript is thoroughly proofread by a native English speaker prior to resubmission, to check for language errors.

A native English speaker at American Journal Experts has proofread the revised paper thoroughly.

Reviewer(s)' Comments to Author:

Reviewer: 2

Reviewer Name: Liane Ginsburg

Institution and Country: York University, Canada

Please state any competing interests or state 'None declared': No competing interests.

Please leave your comments for the authors below

The authors have attended to several concerns raised in the previous review. The strength of this paper is the extensive Chinese sample allowing the authors to contribute to an understanding of PSC perceptions in that jurisdiction. Below I note some concerns about the results – mainly it seems like a number of small, non-significant differences between groups (eg depts., staff groups) are highlighted in the results and discussion and this has the effect of playing down some of the more interesting differences that were found. Concrete instances of this concern are elaborated on below. IT may just be that the results need to be described more clearly and succinctly – it is difficult to tell. More specific things that need to be addressed (in the order they appear in the manuscript) are as follows:

We truly appreciate your valuable comments and suggestions and did our best to revise the paper accordingly.

Abstract – In 19 – not sure what is meant by “should obtain more concerns”

We have deleted this statement (Abstract, line 19).

Several types need to be corrected:

- Abstract results section

We have corrected the results section in the abstract.

- Introduction section, In 8 readmission rates should be followed by a comma not a period

We have revised the punctuation as requested.

- page 5. Ln 18 should say “whether THERE WERE differences by job type...”

We have revised the text as requested (P5, line 18).

- Methods section p5 In 26 – say “includes “fear of blame...” instead of USE “fear of blame

Thank you. We have modified the content according to your comment (P5, line 26).

- Data sources section, In 30 – clarify what et al departments include. And also clarify if you included pediatrics and other less critical departments – these are not listed here on pp7-8 yet you mention in the study introduction that these departments are less studied (and you mention them further on in the paper) – pls make sure pp7-8 description of units in the sample is accurate.

We added all other surveyed departments in the description, including the department of pediatrics (P7, lines 28,30; P8, lines 1-2).

- reference 32 is not included on your reference list

We apologize for this error, which has been corrected.

- clarity of written English needs to be checked on additions that were made to this version of the manuscript

A native English-speaking editor at American Journal Experts has proofread the revised paper thoroughly.

Page 10 – Analysis section – please describe what a heat map is. Also these were included in the HTML version of the uploaded manuscript but they are not referred to or explained at all in the results. We have retained one figure of a heat map indicating the predictive means of each dimension and overall safety climate perceived by the four types of staff among the different working departments. A description of a heat map has been added in the analysis section (P10, lines 12-14).

Variation among clinical departments and job types (pp11-12): it seems to make sense to only highlight differences for departments of job types where the Cis do not overlap. You mention many differences in the text but in the table the groups have overlapping Cis.

We have revised the descriptions in the results and discussion sections (P11, lines 20-30; P12, lines 1-11).

Page 11 (Variations within Clinical departments among Physicians with Various Titles and Other Personnel) – there is very little text here describing a huge number of results presented in a table. The text needs to highlight and describe any statistically and clinically significant differences. Also the text refers readers to figure 3 but the supplementary file with tables has each labeled as Appendices not figures – this needs to be made consistent

Thank you very much. We have added the corresponding description in this part and referred to Figure 1. We have also deleted the related table in the appendices (P13, lines 9-29; P14, lines 1-7).

Your results on the very high proportion of problematic responses related to “fear of blame and punishment” and “fear of shame” seem to be the most jarring and unique – it may help to focus on these results and the implications for safety and safety improvement as this is in many ways a dimension of PSC that reflects the very essence of the construct.

Thank you. We have added more discussion of this issue (P15, lines 5-12).

The Discussion on page 13 of climate variations by dept does not seem to fit the results in Appendix B given the overlapping Cis so I am not clear why these “differences” are highlighted in the discussion. Similarly, you state in the discussion that PPRs by physicians were systematically higher than other staff in the study but my read of Appendix C is that the docs, nurses and techs (all the clinical staff) have overlapping Cis and that the clinical groups scores show more PPRs than the managers and other non-clinicians. Unless I am reading these data incorrectly, these results are similar to what we find in other countries (managers PSC scores are nearly always more positive as many PSC items are about managerial behaviors and priorities!)

Thank you very much. We have revised our description according to the Cis in the table according to your suggestion (P12, lines 20-28).

The data don't seem to support the “Substantial Concerns in the Pediatric Departments” – their results in Appendix seem similar to other depts, except maybe OB which is lower. Is it the Appendix E findings this piece of discussion is based on?

Thank you. We have modified our descriptions in the results and discussion sections to focus more on characteristics of patient safety in child care (P12, lines 4-11; P17, lines 9-12).

The strength of this study is the extensive nature of the sample. Accordingly, I want to see some more discussion of representativeness in the discussion section – of hospital type (eg secondary), hospital size (eg large), of proportion of staff in each clinical dept surveyed and of the job types. Please present some data to show whether the proportions in your respondent group (described on page 10 in the 1st results paragraph) are representative of the proportions in China?

As we stated in the methods, our study data were from the 54 public general hospitals in three regions representing high, middle and low socioeconomic levels and the eastern, central, and western regions of China. However, staff from different working departments were not fully sampled according

their proportions in the departments. Thus, our results might not be generalizable to all hospitals in China, as we have stated in our study limitations.

In addition, we have no data on the proportions of hospital types (hospital level), hospital size (such as bed), staff in each department, or staff job type. We therefore cannot compare our sample proportions with those in China.

However, we analyzed the patient safety climate according to job type and working departments, which may at least partially compensate for these limitations.

Lastly, you state in the response to reviewers that you attended to our suggestion to look at the extent to which staff in different departments “agree” about their safety climate ratings (see Ginsburg & Oore, 2016, BMJQS for a discussion of looking at safety climate strength in addition to climate level which you focus on). You responded that “we have analyzed the variations of the predicted means of perception of patient safety climate on each dimension and overall among physicians with different titles and other personnel within each of 9 departments.” – this is not what climate strength is about and while you certainly do not NEED to performance these analyses – I want to make sure that your rationale for providing variations in predicted means is clear and what you intended to do.

Thank you very much. We have carefully studied the paper you mentioned and read some relevant references on patient safety climate strength, which provided a great deal of enlightenment. We thought it would make sense to further analyze climate strength and distribution. Because of the space limitations of this manuscript, we think it might be better to focus on this issue in another paper. We apologize that we did not respond to the reviewer correctly last time.

Reviewer: 1

Reviewer Name: Raymond Pong

Institution and Country: Centre for Rural and Northern Health Research, Laurentian University, Canada

Please state any competing interests or state ‘None declared’: None

Please leave your comments for the authors below

Having read the revised manuscript, I believe the authors have addressed most of the concerns I raised in my earlier review. I will let the other reviewers decide whether revisions made have adequately addressed their concerns.

However, I should point out that there are still some language deficiencies (e.g., grammatical errors and problematic expressions) in the manuscript. The authors mentioned that they had used the services of a commercial editing agency to help improve the writing. But, the agency did NOT do a very good job. Here are a few examples of problematic writing:

* "We specifically explore... (3) whether differences by job type within the selected clinical departments." [suggested changes: "We specifically explore... (3) whether there were differences by job type in the selected clinical departments."]

Thank you. We have corrected the content (P5, line 18).

* "The clinical auxiliary departments are referred to laboratory department, image department..." [suggested changes: " 'Clinical auxiliary departments' refers to laboratory department, imaging department..."]

Thank you. We have modified the text as suggested (P8, line 2).

* "In this survey, the result revealed that the overall perception of patient climate was relatively good (PPR=9%)." [suggested changes: "The survey results revealed that the perception of overall safety climate was relatively good (PPR=9%)."]

Thank you. We have revised the text (P14, line 22).

* "Besides, the results revealed that the perceptions by managers were relatively consistent with frontline workers on many dimensions, which was different with many previous researches."

[suggested changes: "Besides, the results revealed that the perceptions of managers were relatively consistent with those of frontline workers on many dimensions, but were different than those reported in many previous studies."]

Thank you. We have revised the text (P16, lines 26-28).

There are other similar language problems. I would leave it to the editor to determine whether such minor linguistic blemishes are serious enough to tarnish an otherwise substantial, well-conducted, and worthwhile study and whether some additional effort to clean up the language is needed.

Thank you. A native English-speaking editor at American Journal Experts has proofread the revised paper thoroughly.

I also would like to make another language-related suggestion. When referring to departments within a hospital, the authors used various terms: "clinical departments," "work departments," and "clinical departments." I think some consistency in terminology would be helpful and suggest they use the term "clinical departments" throughout the manuscript.

Thank you. The departments in our study included not only clinical departments but also administrative offices, which were categorized as "other" departments. We have standardized the term "working departments" throughout the manuscript.

VERSION 3 – REVIEW

REVIEWER	Raymond Pong Laurentian University Canada
REVIEW RETURNED	27-Sep-2017

GENERAL COMMENTS	This is the third time I review this manuscript. It is my opinion that two rounds of revision by the authors based on comments and suggestions provided by reviewers (including me) have improved the manuscript considerably. The substantive and interpretation issues raised in my first review have mostly been adequately addressed. I pointed out language and expression problems in my second review. It appears that the authors have done a good job cleaning up the writing, though a few minor problems still exist, such as: (1) Mostly in the Introduction section, "patient safety climate" is inevitably preceded by "the". But, in most cases, the definite article is not needed or sounds awkward. (2) Last paragraph in the Data Source section: The second sentence -- "Written consent ecause our survey of employees was anonymous and had" -- is incomplete, contains typos, and is incomprehensible. Needs rewriting. (3) Second paragraph in the Discussion section: What is "HRO"? "HRO" has not been defined or explained earlier in the manuscript. Regarding statistics, it appears that a specialist statistical review has already been carried out and that the authors have made necessary changes based on the review. Thus, I believe no further statistical review at this stage is needed.
--

REVIEWER	L. Ginsburg York University Canada
REVIEW RETURNED	02-Oct-2017

GENERAL COMMENTS	The strength of this study is the extensive nature of the sample. I continue to have the same concern I had when I reviewed R1 (and I did not see a response to reviewers in the documents that were provided for this R2 review). In summary, many of the results and discussion of climate variations by dept does not seem to fit the results given the overlapping confidence intervals in the tables so I am not clear why groups with overlapping CIs are highlighted as "different" in the discussion. To me, it seems the more important 'comparisons' to be made from these data are for the PPRs between dimensions. Your results on the very high proportion of problematic responses related to "fear of blame and punishment" and "fear of shame" (compared to other dimensions of PS climate) seem to be the most jarring and unique – it may help to focus on these results and the implications for safety and safety improvement as this is in many ways a dimension of PSC that reflects the very essence of the construct. Other things to be noted: P7, In 11 – make this an overall statement about sampling and representativeness. State: "Overall, we recruited a sample of 18 tertiary...". Add a further statement about how representative this sample is of different hospital types, sizes, geographic and socioeconomic areas. This is needed to help international readers not familiar with these details about China. P8, In 6 – this section on data collection procedures has several words missing. It also remains unclear whether the decision to complete a survey or not was anonymous. It is clear that no identifiers were used on the surveys but did the coordinators know whether someone chose to complete a survey? And what is the coordinators role? Are they hospital employees or study personnel from the university? This needs to be clear. These details are important for understanding why response rates were so high. P8 – Psychometric Analysis paragraph – can you please report the Comparative Fit Index (CFI). PP13-14 : Figure 1 is used referred to for all of these results on these 2 pages and it is not legible due to the size of the font . P15, In 24, change "Ors" to "ORs" P15, In 26-27, it would be better to avoid using terms like "significantly high" It is also helpful to avoid describing a PPR as "remarkably high" without saying what this assessment is compared to P17, Ins 10-12: You state that "In this investigation, pediatrics personnel responded with higher PPRs in several dimensions of the safety climate, especially "organizational resources for safety" and "psychological safety."...It is not clear from the results that the pediatrics PPRs on these 2 dimensions are SIGNIFICANTLY higher than other areas given the overlapping confidence intervals. I have made this point in previous reviews. Perhaps I am missing something about what analyses were performed? Please clarify as you don't want to suggest actions based on pediatrics having more PPRs unless the data are clear that this area is indeed scoring lower (rather than just noise in the data) P17, In 16, in discussing the shortage of pediatric doctors as a possible explanation for higher PPRs in this group, it is useful to provide a comparison statistic to other specialities in china, in
---

	addition to the comparison to the Us. For instance, you state that the “number of pediatric doctors per 1,000 children is only 0.5, one-third of the ratio in the US”, what is the number of internists or OBGYNs per 1,000 adults and maternal age women, respectively? P18, ln 6 – the way literature on decreasing cognitive function following long work hours and post call is used suggests this is an issue only for ICUs and ED...is that what you intended to suggest?
--	---

VERSION 3 – AUTHOR RESPONSE

Editor, Journal of BMJ Open
Dec. 9th, 2017

Dear Dr. Emma Gray and Reviewers:

We greatly appreciate the significant attention of the editor and reviewers to the manuscript “Patient Safety Climate in Public General Hospitals in China: Multilevel Modeling of Differences Associated with Department and Job Type” (Manuscript ID bmjopen-2016-015604.R2). We are particularly grateful for your comments on our paper, and we have further revised the manuscript based on those comments. Our point-by-point responses follow. In the manuscript, we have highlighted our changes in yellow or with tracked changes.

Sincerely,

Ping Zhou

Reviewer(s)' Comments to Author:

Reviewer: 1

Reviewer Name: Raymond Pong

Institution and Country: Laurentian University, Canada

Please state any competing interests or state 'None declared': None declared

Please leave your comments for the authors below

This is the third time I review this manuscript. It is my opinion that two rounds of revision by the authors based on comments and suggestions provided by reviewers (including me) have improved the manuscript considerably.

Thank you very much for your comments.

The substantive and interpretation issues raised in my first review have mostly been adequately addressed. I pointed out language and expression problems in my second review. It appears that the authors have done a good job cleaning up the writing, though a few minor problems still exist, such as:

(1) Mostly in the Introduction section, "patient safety climate" is inevitably preceded by "the". But, in most cases, the definite article is not needed or sounds awkward.

We have deleted “the” in front of “patient safety climate” in many cases in the introduction section based on your suggestion.

(2) Last paragraph in the Data Source section: The second sentence -- "Written consent because our survey of employees was anonymous and had" -- is incomplete, contains typos, and is incomprehensible. Needs rewriting.

We are sorry for our mistake. We have rewritten this sentence (P8, lines 6-8).

(3) Second paragraph in the Discussion section: What is "HRO"? "HRO" has not been defined or explained earlier in the manuscript.

We added the definition of HRO based on your suggestion (P14, line 17).

Regarding statistics, it appears that a specialist statistical review has already been carried out and that the authors have made necessary changes based on the review. Thus, I believe no further statistical review at this stage is needed.

Reviewer: 2

Reviewer Name: L. Ginsburg

Institution and Country: York University, Canada

Please state any competing interests or state 'None declared': None declared

Please leave your comments for the authors below

The strength of this study is the extensive nature of the sample. I continue to have the same concern I had when I reviewed R1 (and I did not see a response to reviewers in the documents that were provided for this R2 review). In summary, many of the results and discussion of climate variations by dept does not seem to fit the results given the overlapping confidence intervals in the tables so I am not clear why groups with overlapping CIs are highlighted as “different” in the discussion. To me, it seems the more important ‘comparisons’ to be made from these data are for the PPRs between dimensions. Your results on the very high proportion of problematic responses related to “fear of blame and punishment” and “fear of shame” (compared to other dimensions of PS climate) seem to be the most jarring and unique – it may help to focus on these results and the implications for safety and safety improvement as this is in many ways a dimension of PSC that reflects the very essence of the construct.

Thank you for this comment. We have checked the CIs and modified the corresponding description and discussion (P11 line 24-P12 line 12; P12 line 21-P13 line 29; P15 lines 5-22; P17 lines 2-5).

We also recognized the strength of the PSCHO tool and have noted in the "survey instrument" part of the methods section (P5, lines 23-26) that it included “fear of blame” and "fear of shame" to measure potential barriers to improving patient safety, both of which capture underlying characteristics of Chinese culture.

Other things to be noted:

P7, In 11 – make this an overall statement about sampling and representativeness. State: “Overall, we recruited a sample of 18 tertiary...”. Add a further statement about how representative this sample is of different hospital types, sizes, geographic and socioeconomic areas. This is needed to help international readers not familiar with these details about China.

Thank you for this comment. We have added more information about our hospital sample in that section to indicate the representativeness of the hospitals (P7 lines 8-12).

P8, In 6 – this section on data collection procedures has several words missing. It also remains unclear whether the decision to complete a survey or not was anonymous. It is clear that no identifiers were used on the surveys but did the coordinators know whether someone chose to complete a survey? And what is the coordinators role? Are they hospital employees or study personnel from the

university? This needs to be clear. These details are important for understanding why response rates were so high.

P8 – Psychometric Analysis paragraph – can you please report the Comparative Fit Index (CFI). During the survey, the questionnaires were distributed and collected according to our study design by trained coordinators who were employees in the surveyed hospitals or administrators from the local health bureaus. Although the questionnaires were self-reported and anonymous, the coordinators knew how many questionnaires were distributed to the staff in their surveyed hospitals and departments. If there were fewer questionnaires returned to the coordinators, they returned to the surveyed hospitals or departments to remind those who were willing to participate to return their questionnaires.

We have added some of this information to the paper (P8 lines 9-12).

The Comparative Fit index (CFI) was 0.913 (P8 line 26).

PP13-14 : Figure 1 is used referred to for all of these results on these 2 pages and it is not legible due to the size of the font .

The original Figure 1 (TIFF of 3.2MB) is clear with proper font size. We submitted it individually.

However, after we submitted it, the figure was merged into the PDF, and the Figure 1 in the version of the PDF that you saw was no longer legible. We believe that the publisher can address this issue.

P15, In 24, change “Ors” to “ORs”

Thank you. We have made this change (P15 line 16).

P15, In 26-27, it would be better to avoid using terms like “significantly high” It is also helpful to avoid describing a PPR as “remarkably high” without saying what this assessment is compared to

Thank you. We have modified this text.

P17, Ins 10-12: You state that “In this investigation, pediatrics personnel responded with higher PPRs in several dimensions of the safety climate, especially "organizational resources for safety" and "psychological safety."...It is not clear from the results that the pediatrics PPRs on these 2 dimensions are SIGNIFICANTLY higher than other areas given the overlapping confidence intervals. I have made this point in previous reviews. Perhaps I am missing something about what analyses were performed? Please clarify as you don't want to suggest actions based on pediatrics having more PPRs unless the data are clear that this area is indeed scoring lower (rather than just noise in the data)

Thank you for this comment. We have modified the text in this section of the discussion (P17, lines 3-5).

P17, In 16, in discussing the shortage of pediatric doctors as a possible explanation for higher PPRs in this group, it is useful to provide a comparison statistic to other specialties in china, in addition to the comparison to the Us. For instance, you state that the “number of pediatric doctors per 1,000 children is only 0.5, one-third of the ratio in the US”, what is the number of internists or OBGYNs per 1,000 adults and maternal age women, respectively?

Based on the 2016 China Health Statistics Yearbook and the sixth national population census, we estimated the number of OBGYNs per 1,000 maternal-age women and added this information to the text (P17, lines 10-13).

P18, In 6 – the way literature on decreasing cognitive function following long work hours and post call is used suggests this is an issue only for ICUs and ED...is that what you intended to suggest?

Decreased cognitive function following long work hours is common in various working departments. However, we want to emphasize that the staff working in ICUs and EDs are prone to have a high workload, to work long hours and to face a high level of pressure because these work areas have some intrinsic characteristics such as a higher level of risk, complexity, and difficulty, a faster pace

and lower predictability than other working departments. In addition, we cite the literature on the impacts of burnout or long work hours to indicate their potential bad consequences. Because we perhaps did not express this information clearly, we have modified the text (P18, lines 4-6).